# WEAKLY-SUPERVISED LEARNING OF DISENTANGLED AND INTERPRETABLE SKILLS FOR HIERARCHICAL REINFORCEMENT LEARNING

## ABSTRACT

Hierarchical reinforcement learning (RL) usually requires skills that can be applicable to various downstream tasks. While many recent works have been proposed to learn such skills in supervised or unsupervised manners, the learned skills are often entangled, which makes it difficult to interpret those skills. To alleviate this, we propose a novel WEakly-supervised learning approach for learning Disentangled and Interpretable Skills (WEDIS) from the continuous latent representations of trajectories. We accomplish this by extending a trajectory variational autoencoder (VAE) to impose an inductive bias with weak labels, which explicitly enforces the trajectory representations to be disentangled into factors of interest that we intend the model to learn. Given the latent representations as skills, a skill-based policy network is trained to generate similar trajectories to the learned decoder of the trajectory VAE. Additionally, we propose to train a policy network with single-step transitions and perform the trajectory-level behaviors at test time with the knowledge on the skills, which simplifies the training procedure for the policy. With a sample-efficient planning strategy based on the skills, we show that our method is effective in solving the hierarchical RL problems in experiments on several challenging navigation tasks with a long horizon and sparse rewards.

## 1 INTRODUCTION

Deep reinforcement learning (RL) has achieved great success for various applications, ranging from playing games (Mnih et al., 2013; Silver et al., 2016) to complex locomotion and robots control (Lillicrap et al., 2015; Schulman et al., 2015; 2017; Haarnoja et al., 2017). However, several challenges such as sparse rewards or inadaptability to unlearned tasks still hinder its practical usages in real-world problems. To alleviate these challenges, hierarchical RL (Sutton et al., 1999; Dietterich, 2000) has been studied where an agent pre-learns reusable skills from prior experiences and hierarchically solve higher-level problems by combining the skills. Two issues need to be resolved for the successful deployment of the hierarchical RL; how to learn useful skills and how to effectively make use of the skills for various downstream tasks.

A possible approach for skills that can be applicable to various downstream tasks is to learn without task-specific rewards (Eysenbach et al., 2018). Another way to achieve the useful skills is to make them predictable. To learn those skills, (Co-Reyes et al., 2018; Sharma et al., 2019) proposed to combine model-free and model-based RL approaches, where a skill-based predictive model, a dynamics model over the latent space, is trained together with a skill-based policy network.By using the predictive model for model-based planning during testing time, these works showed to efficiently solve various downstream tasks without the need to learn additional higher-level policies. However, since they did not consider how the skill is embedded into the latent space, the factors consisting of the skill often are entangled when the skill is a continuous latent variable. Compared to the entangled one, the skill consisting of disentangled factors has several advantages in its applicability in that the factors can be separately interpreted and handled.

In this paper, we introduce a novel WEakly-supervised learning approach for learning Disentangled and Interpretable Skills (WEDIS) from the continuous latent representations of trajectories that are composed of several generative factors, e.g., speed, direction, and curvature. To this end, we propose

a weakly-supervised trajectory variational autoencoder (WET-VAE) model that is an extension of the trajectory VAE (Co-Reyes et al., 2018) consisting of a recurrent neural network (RNN). We leverage the weak labels (Margonis et al., 2020) to enforce an inductive bias on the model, which explicitly enforces the trajectory representations to be disentangled into factors of interest that we intend the model to learn. To train the WET-VAE, we first synthetically generate a trajectory dataset by the combination of several factors of interest, because the trajectories obtained by an online exploration are likely to contain meaningless samples such as random walks. With the trajectory dataset, the WET-VAE model is trained apart from a policy network. It is worthy of noting that while this is similar to imitation learning, our data acquisition is much simpler than collecting expert demonstration.

Sequentially, we train a skill-based policy network with the WET-VAE fixed. Given the latent representations as skills, the skill-based policy network is trained to generate similar trajectories with the decoder of the WET-VAE by minimizing the KL divergence between two trajectory distributions. However, training a policy to generate a trajectory given a skill is difficult since it is unlikely to explore the corresponding trajectory in the training procedure. Instead, we propose to train the policy network with the single-step transitions and perform the trajectory-level behaviors in the test time, which can be achieved with the knowledge of the learned skills. This simplifies the training procedure of the policy, and also allows for a sample-efficient large-scale planning strategy with the scaled trajectories. In experiments in Mujoco Ant environment, we show that our disentangled and interpretable skills are effective in solving challenging sparse reward and long-horizon problems in 2D navigation in mazes.

## 2 RELATED WORKS

Numerous approaches (Sutton et al., 1999; Bacon et al., 2017; Florensa et al., 2017; Hausman et al., 2018; Haarnoja et al., 2018; Eysenbach et al., 2018; Shankar et al., 2019; Shankar & Gupta, 2020; Co-Reyes et al., 2018; Sharma et al., 2019) have explored on learning reusable skills in RL to solve challenging long-horizon or sparse reward problems. (Sutton et al., 1999) pioneered a way to control higher-level abstraction by introducing an option-framework, which learns low-level primitives in a top-down manner. (Bacon et al., 2017) proposed an option-critic architecture that learns sub-policies of options. Also, several works (Florensa et al., 2017; Hausman et al., 2018; Haarnoja et al., 2018) introduced to learn skills with multiple tasks in a bottom-up manner. However, designing reward functions still requires expert knowledge and such task-specific rewards may limit a generalization ability of the agent to the downstream tasks. To overcome this issue, recent works (Eysenbach et al., 2018; Achiam et al., 2018; Co-Reyes et al., 2018; Sharma et al., 2019; Campos et al., 2020) proposed an unsupervised framework that does not require a hand-specified reward function.

Model-based RL methods (Levine et al., 2016; Nagabandi et al., 2018; Chua et al., 2018; Ha & Schmidhuber, 2018) aim to learn a dynamics model of the environment. While these works are capable of solving unlearned tasks without the needs of an additional learning via planning through the dynamics model, they are often at the risk of falling into over-fitting due to a huge capacity of the required data to explore the environment. Instead of learning the underlying dynamics, some methods (Co-Reyes et al., 2018; Sharma et al., 2019) attempted to combine the model-free and model-based RL for learning a skill-based predictive model and a skill-based policy. Despite the improved results, they still suffer from the lack of the interpretability of the skills.

Learning disentangled latent representations of factors of variation within dataset is beneficial to a variety of downstream tasks such as few-shot classification and data generation, thanks to the interpretability of the disentangled factors. (Higgins et al., 2016) proposed $\beta$-VAE, an unsupervised method to learn the disentangled representations by modifying the weight of the KL-divergence term of the VAE (Kingma & Welling, 2013; Rezende et al., 2014) greater than one. Afterwards, while several variants (Kim & Mnih, 2018; Chen et al., 2018) improved the $\beta$-VAE by introducing a total correlation (TC) term, (Locatello et al., 2019a) pointed out the inherent limitation of the purely unsupervised approaches and emphasized the need of an inductive bias. Recent works (Locatello et al., 2019b; Shu et al., 2019; Locatello et al., 2020; Margonis et al., 2020) proposed various forms of weak supervision to encourage the inductive bias to learn the disentangled representations. While there are various categories on the weak labels, we used them in terms of ones that 1) are roughly divided into fewer classes and 2) can be obtained with programming by using the knowledge on the factors without the need for manual labeling.

## 3 PRELIMINARIES

Consider a Markov decision process (MDP) $(\mathcal{S}, \mathcal{A}, \mathcal{P}, r, \rho_0, \gamma)$, where $\mathcal{S}$ is a set of states, $\mathcal{A}$ is a set of action, $\mathcal{P} : \mathcal{S} \times \mathcal{A} \times \mathcal{S} \to \mathbb{R}^+$ is a transition probability distribution, $r : \mathcal{S} \times \mathcal{A} \to \mathbb{R}$ is a reward function, $\rho_0 : \mathcal{S} \to \mathbb{R}^+$ is an initial state distribution and $\gamma \in (0, 1)$ is a discount factor. We denote a stochastic policy as $\pi : \mathcal{S} \times \mathcal{A} \to \mathbb{R}^+$. RL has a goal of maximizing the expected discounted sum of rewards for an episode horizon $H_E$:

$$\eta(\pi) = \mathbb{E}_{\pi}[\sum_{t=0}^{H_E} \gamma^t r(s_t, a_t)] \tag{1}$$

Variational autoencoder (VAE) optimizes variational the lower bound of the marginal likelihood of dataset. Given an observed datapoint $x$, the variational lowerbound is defined as :

$$\log p_\theta(x) \geq L(\theta, \phi\,; x) = \mathbb{E}_{q_\phi(z|x)}[\log p_\theta(x|z)] - D_{KL}(q_\phi(z|x)\|p(z)), \tag{2}$$

where $p(z)$ is a prior distribution of a latent variable $z$, the decoder $p_\theta(x|z)$ is a generative model given a latent $z$ parameterized by $\theta$, and the encoder $q_\phi(z|x)$ is an approximate posterior distribution parameterized by $\phi$. In Equation 2, the first term is the reconstruction term of the autoencoder, and the second term is the KL divergence regularization. In our work, we will focus on the aspect of the generative model of the decoder.

## 4 WEAKLY SUPERVISED LEARNING OF DISENTANGLED AND INTERPRETABLE SKILL (WEDIS)

Our framework consists of three stages; 1) generating trajectory training data with factors of interest 2) training the WET-VAE model, whose decoder is used for the predictive model and 3) training a policy network to generate the similar trajectories with the predictive model conditioned on skills. The generation process of the trajectory dataset is explained in Appendix A.1.1 due to the lack of space. As a notation, we will use superscript for factors and subscript for time steps. The WEDIS algorithm is summarized in Figure 2.

### 4.1 LEARNING DISENTANGLED AND INTERPRETABLE REPRESENTATIONS OF TRAJECTORY

To learn the temporally extended behaviors, (Co-Reyes et al., 2018) proposed a trajectory VAE model consisting of the RNN architecture. The trajectory VAE learns latent representations of trajectories, which will be used as skills for a policy. However, this model, which learns the representations in the unsupervised manner, does not consider which factors of variation of a trajectory are embedded in the latent space. Thus, the factors that are often entangled make the interpretation of the representations difficult, exposing limitations in further applicability of the learned skills. To address this, we propose a weakly-supervised trajectory VAE (WET-VAE) model that leverages an inductive bias in the form of weak supervision (Margonis et al., 2020) to explicitly enforce the model to learn the disentangled representations consisting of desired factors, yielding interpretable skills.

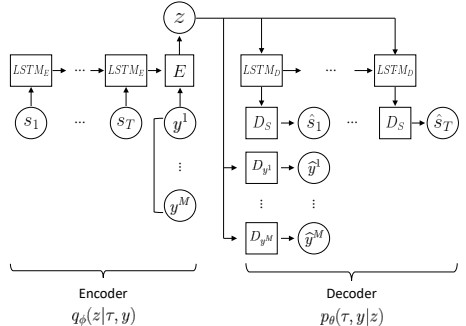

Figure 1: Computation graph for the WET-VAE model. For weak supervision, we add a set of weak labels $\{y^1, ..., y^M\}$ to the model. These labels encourage the model to learn a disentangled trajectory representation $z$ that consists of corresponding factors. We assume the fixed initial state $s_0$ at the origin.

Consider a latent-variable generative model $p(\tau|z)$ to generate a trajectory $\tau$ given a latent variable $z$. We assume the fixed initial state $s_0$ at the origin as when given other initial states we can obtain the next states with a linear translation based on the initial states such that $p(s|s_0, z) = p(s - s_0|z)$. Considering $M$ factors of interest to generate trajectories, the weak supervision can be provided by simply adding a set of $M$ weak labels $y = \{y^1, ..., y^M\}$ to the generative model, where each label $y^m$ is one-hot encoded vector

for each factor. The idea is that a latent representation $z \in \mathbb{R}^M$, which can generate the trajectories based on the $M$ disentangled generative factors, should also be able to reconstruct the factors. Assuming that the trajectory and the factors that are represented as the multiple labels satisfy conditional independence with respect to a given $z$, the generative model is extended with the labels $p(\tau, y|z) = p(\tau|z)p(y^1|z)\cdots p(y^M|z)$. Then, the variational lower bound of the marginal joint distribution $p(\tau, y)$ can be formulated as follows:

$$L(\theta, \phi\,; \tau, y) = \mathbb{E}_{q_\phi(z|\tau,y)}[\log p_\theta(\tau, y|z)] - D_{KL}(q_\phi(z|\tau, y)\|p(z))$$

$$= \mathbb{E}_{q_\phi(z|\tau,y)}[\sum_{t=1}^{T} \log p_\theta(s_t|s_{1:t-1}, z) + \sum_{m=1}^{M} \log p_\theta(y^m|z)] - D_{KL}(q_\phi(z|\tau, y)\|p(z)),$$

(3)

where $p_\theta(\tau|z) = p_\theta(s_1|z)p_\theta(s_2|s_1, z)\cdots p_\theta(s_T|s_1, s_2, ..., s_{T-1}, z)$. Since $p_\theta(y^m|z)$ can be understood as a classifier for each factor, the factors should be *distinctly* embedded in a latent representation $z$ for high classification probability. As a result, this enforces a disentangled representation of the factors.

Practically, the scales of the values of the log-likelihoods of the states and labels are different due to the difference in dimensionality. To fill the gap, we introduce a balancing weight $\gamma$ inspired by (Margonis et al., 2020). We also use a weight $\beta > 1$ to emphasize the KL divergence term for better disentanglement in the spirit of the $\beta$-VAE (Higgins et al., 2016). Then, the final objective function becomes:

$$L(\theta, \phi\,; \tau, y, \beta, \gamma)$$
$$= \mathbb{E}_{q_\phi(z|\tau,y)}[\sum_{t=1}^{T} \log p_\theta(s_t|s_{1:t-1}, z) + \gamma \cdot \sum_{m=1}^{M} \log p_\theta(y^m|z)] - \beta \cdot D_{KL}(q_\phi(z|\tau, y)\|p(z))$$

(4)

The WET-VAE model is trained to maximize Equation 4. To handle the sequential data, we use the RNN architecture with LSTMs as in Figure 1. With the support of the weak supervision, this model can learn the disentangled representations of trajectories that consist of factors of variation contributing over different time steps.

## 4.2 LEARNING POLICY WITH LEARNED SKILLS

### 4.2.1 OPTIMIZATION OF POLICY NETWORK TO IMITATE PREDICTIVE MODEL

After training of the WET-VAE, its decoder $p_\theta(\tau|z)$ is used as a skill-based predictive model of trajectories. To learn the disentangled and interpretable skills, the policy network is trained to generate similar trajectories with the predictive model based on the skills. It can be accomplished by optimizing the policy to minimize the KL divergence between the distributions of trajectories generated by the two networks.

$$\min_{\psi} \quad \mathbb{E}_{p(z)}[D_{KL}(p_\psi(\tau|z)\|p_\theta(\tau|z))],$$

(5)

where $\psi$ is a parameter of the skill-based policy network $\pi_\psi(a|\tau, z)$ and $p_\psi(\tau|z)$ is the distribution of the trajectories generated by the policy interacting with the environment, which can be computed as $p_\psi(\tau|z) = \int \cdots \int_{a \in \mathcal{A}}[\prod_{t=1}^{T} p(s_t|s_{t-1}, a_t)\pi_\psi(a_t|s_{0:t-1}, z)]\,da_1\cdots da_T$. Since inferring $p_\psi(\tau|z)$ is intractable due to the lack of knowledge on the transition probability distribution $p(s_t|s_{t-1}, a_t)$ of the underlying dynamics in the environment, Equation 5 cannot be optimized by direct backpropagation with respect to $\psi$. Instead, following (Co-Reyes et al., 2018), we optimize the skill-based policy network with the RL method by rewriting (see Appendix A.2.2) the equation as follows:

$$\max_{\psi} \quad \mathbb{E}_{p_\psi(\tau,z)}[\log p_\theta(\tau|z)] + \mathcal{H}(p_\psi(\tau|z)),$$

(6)

where $p_\psi(\tau, z) = p_\psi(\tau|z)p(z)$ and $\mathcal{H}(p_\psi(\tau|z))$ is the entropy of the trajectories generated by policy given skills. By using the log-likelihood $\log p_\theta(\tau|z)$ calculated by the predictive model for a trajectory explored by the policy as a reward function, we can optimize Equation 6 with conventional model-free RL algorithms (Schulman et al., 2015; 2017; Haarnoja et al., 2017) with entropy regularization. In our implementation, we adopted the soft actor-critic (SAC) (Haarnoja et al., 2017) algorithm that includes the entropy regularization as a part of optimization.

Note that in contrast to (Co-Reyes et al., 2018) that simultaneously optimizes both the trajectory VAE and the policy network with trajectory samples collected from an exploration, our method increases stability in the training since the pre-trained decoder provides reliably fixed rewards.

### 4.2.2 TRAINING POLICY WITH SINGLE-STEP TRANSITIONS

Given a skill, a policy network needs to explore the trajectory similar to the predictive model to receive a reward in Equation 6. However, it is very unlikely to find all trajectories corresponding to each skill by an exploration, which makes the training difficult. To overcome this, we propose to train the policy network with single-step transitions instead of the full trajectories, and perform the trajectory-level behaviors at test time. This can be achieved by exploiting the known knowledge on the factors since the factors of our disentangled skills are interpretable.

Consider a latent $z \in \mathcal{Z} = \left(\mathcal{Z}^{single}, \mathcal{Z}^{multi}\right) \subseteq \mathbb{R}^{M_1+M_2}$ that can be separated to $z^{single} \in \mathcal{Z}^{single} \subseteq \mathbb{R}^{M_1}$ contributing to single-step transition and $z^{multi} \in \mathcal{Z}^{multi} \subseteq \mathbb{R}^{M_2}$ contributing over multiple time steps. For example, while speed and direction can be included in $\mathcal{Z}^{single}$, curvature and acceleration can be included in $\mathcal{Z}^{multi}$. As a trajectory consists of a sequence of single-step transitions, we can generate the same trajectory by a latent $z^{single}$ in combination with $z^{multi}$. That is, with the known relations of the factors, we can compute $z^{single}$ of the next time step, which has the same effect of the execution of the full latent $z$. To this end, we introduce relation functions $f$ to relate the relevant factors. For a $m^{th}$ factor $z^m$ of $z^{single}$, the relation function for the factor can be expressed as $z_{t+1}^m = f^m(z_t^m, z^{multi})$ and properly chosen by their relationships. For instance, we can set a relation function for the speed factor together with the acceleration factor such as $z_{t+1}^{speed} = f^{speed}(z_t^{speed}, z^{acc}) = z_t^{speed} + weight \times z^{acc}$, where the weight can be properly chosen by testing the decoder and policy heuristically after training.

In this way, we train the policy network with $z^{single}$ over single-step transitions in Equation 6, and make use of the full skill $z$ with the relation functions at test time. As the single-step transition does not require to handle the temporal information, we use a feedforward network for the policy instead of a RNN.

---

**Algorithm 1:** WEDIS

**Require**: M generative factors, weak labeling criteria
**Generate**: Trajectory dataset by combination of the
    factors $\mathcal{D} \leftarrow \{\tau\}_{n=1}^N = \{s_0, ..., s_T\}_{n=1}^N$
**Initialize**: parameters $\phi$ (encoder), $\theta$ (decoder),
        $\psi$ (actor), $\bar{\psi}$ (critic), $\hat{\psi}$ (value function)
**while** *Training* **do**     // WET-VAE training
    Sample trajectory batch $\mathcal{B} \sim \mathcal{D}$
    Compute labels $\{y^1, ..., y^M\}_\mathcal{B}$
    Optimize $\phi$ and $\theta$ by Equation 4
**end**
**Test**: Check factors in $z$
**while** *Training* **do**     // policy training
    Sample $z^{single}$ from prior
    Set $z = (z^{single}, z^{multi})$ where $z^{multi} = (0, ..., 0)$
    Execute actions $a \sim \pi_\psi(a|s, z^{single})$
    Collect the samples in replay buffer $\mathcal{R}$
    Sample transition batch $\{s_t, a_t, s_{t+1}\}_\mathcal{B} \sim \mathcal{R}$
    Compute reward $r = \log p_\theta(s_{t+1}|s_t, z)$
    Optimize $\psi$, $\bar{\psi}$ and $\hat{\psi}$ with the SAC by Equation 6
**end**
**Test**: Find relation functions for each factor in $z^{single}$

Figure 2: Illustration of WEDIS algorithm. Two networks are trained separately. After the training procedure of the two networks, we check the embedded factors by the latent traversal and find the relation functions by heuristically testing the WET-VAE decoder and policy.

## 5 PLANNING WITH TRAJECTORY SCALING

---

**Algorithm 2:** MPC with trajectory scaling

---

**Require**: predictive model $p_\theta(\tau|s, z)$, policy $\pi_\psi(a|s, z^{single})$,
  M relation functions $f^m$ for each $z^m$ in $z^{single}$, reward function,
  episode horizon $H_E$, primitive horizon $H_z$, trajectory length $T$,
  planning horizon $H_P$, initial state $s_0$, sample size $N$
**for** $i \leftarrow 1$ **to** $H_E/(H_z \times T)$ **do**
    $\hat{s}_0 \leftarrow s_0$
    Sample latent sequences $\{z_1, .., z_{H_P}\}_{n=1}^N$ from a distribution
    **for** $j \leftarrow 1$ **to** $H_P$ **do**    // for all N samples
       $\tau_j = \{s_1, ..., s_T\} \sim p_\theta(\tau|\hat{s}_0, z_j)$
       Compute distances between the states, $\{\Delta_1, ..., \Delta_T\}$
       $\tau_j^{H_z} = \{s_0 + H_z \cdot \Delta_1, ..., s_0 + H_z \cdot (\Delta_1 + \cdots + \Delta_T)\}$
       $\hat{s}_0 \leftarrow s_T^{H_z}$    // the last state of $\tau_j^{H_z}$
    **end**
    Evaluate the rewards of the scaled trajectory sequences
    Choose the first latent $z_1^*$ of the best trajectory
    $(z^{single}, z^{multi}) \leftarrow z_1^*$
    **for** $t \leftarrow 1$ **to** $T$ **do**
       **for** $k \leftarrow 1$ **to** $H_z$ **do**
          Execute action $a \sim \pi_\psi(a|s_{(t-1)\cdot H_z+(k-1)}, z^{single})$
       **end**
       $\{z^m\}_{m=1}^{M_1} \leftarrow \{f^m(z^m, z^{multi})\}_{m=1}^{M_1}$
       $z^{single} \leftarrow (z^m)_{m=1}^{M_1}$
    **end**
    $s_0 \leftarrow s_{T \cdot H_z}$
**end**

---

(a) Predicted trajectory for $T=3$

(b) Scaled trajectory for planning

(c) Actual trajectory

Figure 3: Illustration of trajectory scaling. This trajectory scaling provides large scale planning that is beneficial to solving long-horizon problems.

After the sequential training of the predictive model and policy network, we can solve hierarchical RL problems without any additional learning at test time via model-based planning. For the model-based planning, we employ the model-predictive control (MPC) method (Garcia et al., 1989). With a dynamics model $p_\theta(s_{t+1} \mid s_t, a)$ of the environment, the MPC planner generates several numbers of trajectory samples $\tau = \{s_0, a_1, ..., s_{H_P}\}$ for a finite planning horizon $H_P$, and then evaluates each trajectory according a reward function given for a task. After choosing the first action of the best trajectory maximizing the planning rewards, the agent executes the chosen action and the MPC planner repeats iteratively this procedure on the next state until the episode horizon $H_E$. Using the predictive model $p_\theta(\tau \mid z)$, we can follow the MPC strategy over the latent space $\mathcal{Z}$ instead of the action space $\mathcal{A}$. In this strategy, we first sample sequences of latents $\{z_1, z_2, ..., z_{H_P}\}$ and evaluate the latent sequences based on rewards of the trajectories generated by the predictive model given the latents, where each latent generates sub-trajectory of length $T$.

Even though we can perform the trajectory-level planning based on this predictive model, the scale of the movement lengths from actions of an agent is limited due to its inherent design specification, which generates trivial trajectories with small scales. To perform planning with a meaningful scale of trajectories, we propose a trajectory scaling method. In contrast to a RNN policy that processes temporal information at every time step, our feedforward policy network can consistently perform the single-step transition given a skill $z^{single}$ from $z$. By using this, we hold on $z^{single}$ for $H_z$ steps and change it by the relation functions at every $H_z^{th}$ step, which generates scaled trajectories by $H_z$ times. In the case of the predicted trajectories $p_\theta(\tau|z)$ that are simulated for planning, we can simply scale the trajectories by scaling each transition distance. This trajectory scaling method provides a large-scale planning strategy that can plan $H_P \times H_z \times T$ steps and execute actions over $H_z \times T$ steps for each skill, which is beneficial to solving long-horizon problems with high sample-efficiency by reducing the maximum planning horizon to $H_E/(H_z \times T)$. Figure 3 summarizes the MPC procedure with trajectory scaling.

## 6 EXPERIMENTS

In this section, we first provide training details and qualitative results of the WET-VAE and the policy. Then, we test our algorithm on goal navigation tasks in mazes in the Mujoco (Todorov et al. (2012); Brockman et al. (2016)) Ant environment (Figure 14), which are challenging hierarchical RL problems. For details of the generation process of the trajectory dataset and labeling criteria, please see Appendix A.1.1. The videos of the learned skills and maze navigation are available at https://sites.google.com/view/iclr2022-wedis.

### 6.1 QUALITATIVE RESULTS

#### 6.1.1 TRAINING OF THE WET-VAE NETWORK

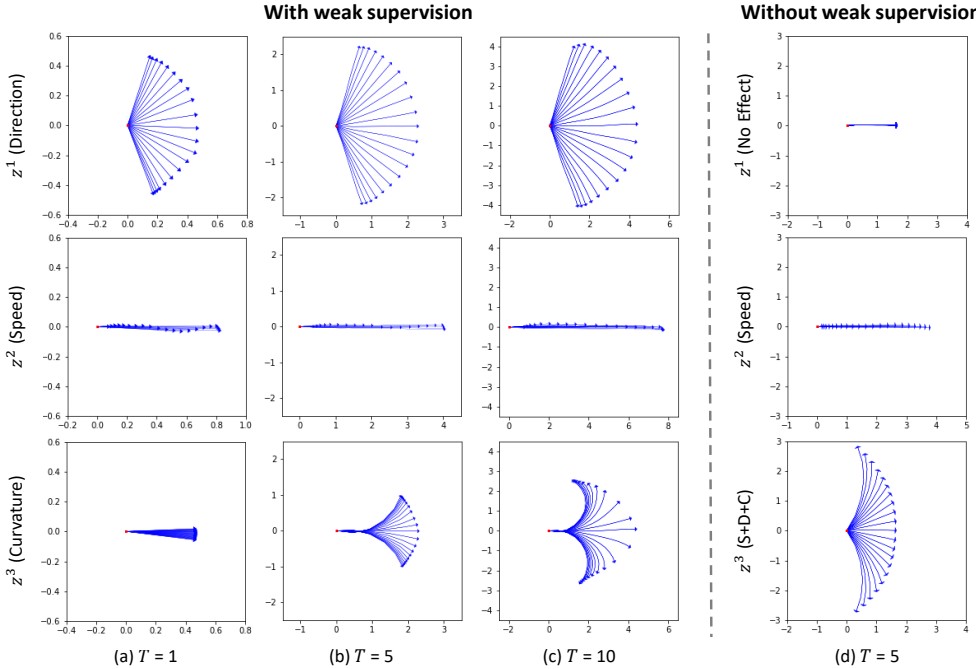

Figure 4: The latent traversals for each skill. (a) $\sim$ (c) show the results of the WET-VAE model, and (d) shows the results the trajectory VAE that does not use any weak supervision, where both models are trained on trajectories with $T = 5$. Each row represents the trajectories generated by latent variables changing values of corresponding dimensions of the latent variable from -1.5 to 1.5, while keeping the others to zeros.

Using the trajectory dataset generated with $T = 5$, we first trained the WET-VAE with a three-dimensional continuous latent variable $z \in \mathbb{R}^3$. We set the prior distribution of the latent variable as an isotropic unit Gaussian distribution $p(z) = \mathcal{N}(0, 1)$. Also, for the posterior, we set the distribution as a Gaussian distribution with a diagonal covariance matrix.

To demonstrate the benefit of the weak supervision in training the WET-VAE, we plot the latent traversals of trajectories of the decoder in Figure 4. For each traversal, the latents are sampled by changing values of one dimension from -1.5 to 1.5, while keeping values of the other dimensions to zeros. Figure 4a shows single-step transitions based on each factor. While $z^1$ and $z^2$ clearly capture the factors on the direction and speed, it looks like that $z^3$ has almost no effect. Since $z^3$ involves the factor of the curvature, this result is reasonable. On the other hand, it is shown that $z^3$ controls the curvature of trajectories when the trajectory is generated with $T = 5$ in Figure 4b. These results imply that the three factors are suitably disentangled according to each dimension of the latent $z$. Furthermore, even when the trajectory length for generation becomes greater than the length of the training data, the contributions of the learned factors consistently remain as shown in Figure 4c with $T = 10$. Note that the weak supervision can successfully disentangle factors that contribute over

different time steps, where the factors of speed and direction affect at the each single step, and the curvature factor affects across the multiple time-steps.

On the other hand, Figure 4d shows that some factors of skills learned from the trajectory VAE, which is trained without weak supervision, are entangled. While the speed factor seems to be disentangled in the second dimension $z^2$, all factors seems to be still entangled in the third dimension $z^3$. Moreover, no factor is embedded in the first dimension $z^1$ as shown in the first row, implying that this element has no effect on the generation of the trajectories.

### 6.1.2 TRAINING OF THE POLICY NETWORK

The policy network is trained in the Mujoco Ant environment with the rewards calculated by the WET-VAE decoder, acting as the predictive model. Since the latent $z$ consists of the factors of speed, direction and curvature, it is separated to $z^{single} = (z^{speed}, z^{dir})$ and $z^{multi} = z^{cur}$. Thus in Equation 6, the policy is conditioned on $z^{single}$ and the predictive model is conditioned on $z = (z^{single}, 0)$ to provide log-likelihoods for $T = 1$. Also, instead of the full states of the agent, we restrict the states to compute the rewards to two dimensional $x - y$ coordinates (Sharma et al., 2019), which enables the policy to be trained with the predictive model trained with 2D trajectories. For the relation functions, we set an identity function as $f^{speed}(z_t^{speed}, z^{cur})$ for speed and $f^{dir}(z_t^{dir}, z^{cur}) = z_t^{dir} - 0.2 \cdot z^{cur}$ for direction, where we found the weight by heuristic test after training both networks.

In Figure 5, we plot latent traversals of trajectories based on each skill, where each row show the trajectories generated by the predictive model and the policy network, respectively. We plot the speed and direction trajectories with $H_Z = 100$ and $T = 1$, and the curvature trajectory with $H_Z = 40$ and $T = 5$. We compare our algorithm against unsupervised skill learning methods (SeCTAR (Co-Reyes et al., 2018), DADS (Sharma et al., 2019)), which are also model-based methods over the latent space. We set $H_Z = 100$ for DADS, and $T = 50$ for SeCTAR. According to the algorithmic design, DADS fixed a trajectory length $T = 1$, and SeCTAR fixed a horizon $H_Z = 1$. For WEDIS, given each skill, it is shown that the actual trajectories are similar to the corresponding predictive trajectories, which implies that our policy acquired disentangled and interpretable skills. On the other hand, for SeCTAR and DADS, the interpretation of each dimension of the skill is unclear.

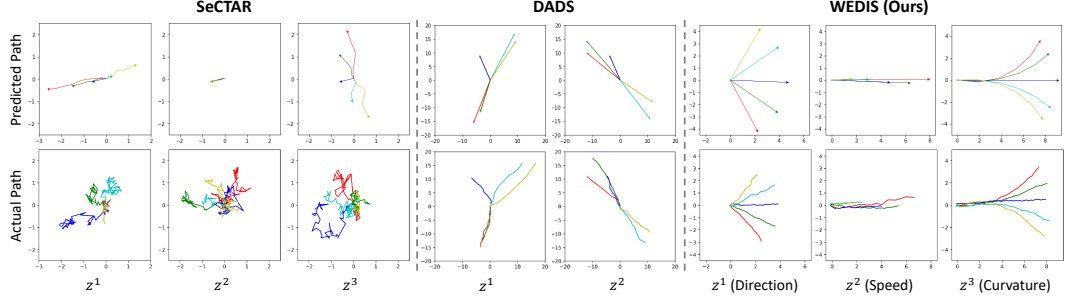

Figure 5: The latent traversals of the decoder for each skill. (Top) Predicted trajectories by the predictive model. (Bottom) Actual trajectories of the agent by the policy network. The colors of (red, green, blue, cyan, yellow) correspond to the values of (-1, -0.5, 0, 0.5, 1) of latents, respectively.

### 6.2 QUANTITATIVE RESULTS

To evaluate the performance on the hierarchical RL problems, we set two challenging maze navigation tasks as shown in Figure 6, which cannot be solved with a single skill, but require to appropriately combine several skills. Also, due to the existence of traps that are close to the goals to some extent, these problems require long-horizon reasoning even for dense rewards cases. In addition to SeCTAR and DADS, we compare with two ablations, single-step

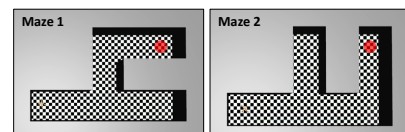

Figure 6: Two navigation tasks. The goals are denoted as the red circles.

transition (WEDIS-S, $T = 1$) and no trajectory scaling (WEDIS-NTS, $H_z = 1$). All of these algorithms can solve the hierarchical RL problems with zero-shot via the MPC over the latent space. The

task reward is given by negative distance to goal position $r(s) = -\|s - g\|_2$. With the task reward and an additional collision penalty, we set the planning reward as $r_d(s) = r(s) - w_p \cdot \mathbb{1}_{(s=collision)}$ for dense reward and $r_s(s) = w_r \cdot \mathbb{1}_{(|r(s)|<\epsilon)} - w_p \cdot \mathbb{1}_{(s=collision)}$ for sparse reward, where $\mathbb{1}$ is an indicator function that outputs one if the condition is true and zero otherwise. $w_r$ and $w_p$ are a reward weight and a penalty weight, respectively, and $\epsilon$ is an acceptable error.

### 6.2.1 NAVIGATION VIA MODEL-BASED PLANNING

In the MPC planning, we evaluated the performance on both dense reward and sparse reward cases. As an evaluation metric, we used the averaged distance between the goal and the final position over ten trials of planning. In Figure 7, we plot the evaluation results according to sample size used for the MPC for comparing the sample-efficiency. WEDIS shows the outstanding results for all the cases, even with less planning samples. Especially, the results on the sparse reward demonstrate that WEDIS is effective in solving long-horizon problems. WEDIS-S and DADS are superior than WEDIS-NTS, implying a larger scale is required for an effective planning, despite of the benefit of the trajectory-level prediction. In addition, while both perform temporally-extended behaviors over single-step transitions, WEDIS-S outperforms DADS, which implies benefits of the disentanglement. For example, the controllability of speed can be beneficial to avoiding collisions to the wall. For SeCTAR, it has poor performance since the skills do not follow predictions well. Some examples of the MPC results are shown in Figure 8, where WEDIS plots smooth trajectories over large scales. For the more details of the planning parameters, please see Appendix A.1.4.

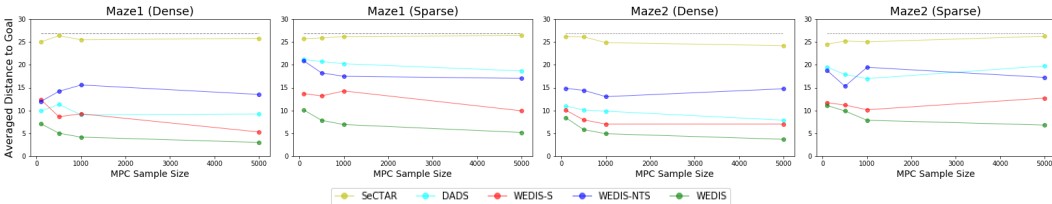

Figure 7: The evaluation results of the MPC on the navigation problems. The dashed gray lines denote goal-distances from the origin. WEDIS outperforms the unsupervised skill learning methods.

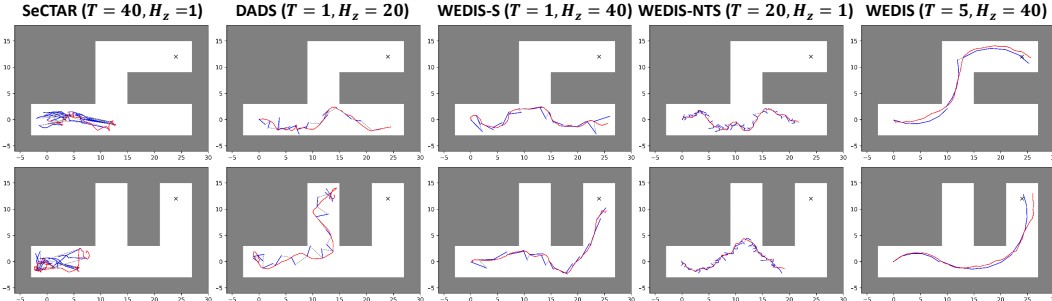

Figure 8: Examples of the MPC results on sparse rewards (Top) Maze 1. (Bottom) Maze 2. The blue and red lines represent the predictive paths and the actual paths, respectively, and the cross marks indicate the goal positions. The dashed gray lines denote error between the last positions of predictive and actual trajectories at each planning step.

## 7 CONCLUSION

This work has introduced a method to learn continuous skills from the disentangled and interpretable representations of trajectories. To do that, we proposed a WET-VAE model by extending the trajectory VAE with weak labels. Using the interpretability, we proposed to train the policy network with single-step transitions and perform the trajectory-level behaviors at test time, which simplifies the exploration problem and provides an effective large-scale planning strategy. In the experiments of challenging navigation tasks, we demonstrated that our method outperforms the unsupervised skill learning methods with higher sample-efficiency.

## 8 REPRODUCIBILITY STATEMENT

For the reproducibility of the proposed method, we will open our code at https://sites.google.com/view/iclr2022-wedis. We also provide the full hyperparameters in Appendix A.1.

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

## A  APPENDIX

### A.1  IMPLEMENTATION DETAILS

#### A.1.1  TRAJECTORY DATASET AND WEAK LABELS

To encourage the disentangled representation learning, we created a trajectory dataset that consists of one-hundred thousand numbers of synthetically generated trajectories. Each trajectory is generated from the combination of several generative factors of interest that we intend the model to learn. For 2D navigation, the trajectories consist of sequences of two-dimensional data points of $x$ and $y$ coordinates. As the factors, we set three factors of speed, direction, and curvature of the trajectories. For the speed, we limited the length of each single-step transition up to one, maintaining it consistently over multiple steps. To allow for only the positive speed and direction, we limited the direction ranging the angles of a right-sided semicircle. For the curvature, we rotated the paths with an angle ranging from -30 to 30 degrees at each transition step. Also, to include straight lines together, we set the angle for curvature to zero with 0.2 probability. Finally, we set the trajectory length $T = 5$, which means a sequence of five transition steps that are composed of six points including an initial point. The details of the factors to generate trajectories used in the main experiments and additional experiments in the appendices are summarized in Table 1.

| Factors | Sampling Distribution |
|---|---|
| Speed | $U(-1, 1)$ (1D) , $U(0, 1)$ (2D) |
| Direction | $U(-\pi/2, \pi 2)$ (semi) , $U(-\pi, \pi)$ (full) |
| Curvature | $U(-\pi/6, \pi/6)$ |
| Acceleration | $U(-s/5, s/5)$, $s$ = sampled speed |

Table 1: The generative Factors

While the training proceeds, weak labels are assigned to the batches as one-hot encoded vectors for each factor, which are computed by programming based on the knowledge of the factors. We roughly define the labeling criteria to divide the factors (speed, direction, and curvature) into four (0.25 interval), four (45 degree interval), and three (straight, right, left) segments. For instance, given a trajectory with (speed = 0.1, direction = 40 degree, curvature angle = -10 degree), then the trajectory is assigned with label values of one, two, and three for each factor.

#### A.1.2  WET-VAE

We used one hidden layer with size of $(64, 64)$ for all units in Figure 1. At each transition step, all of the state and one-hot encoded multiple labels are concatenated as an input. To consider state-agnostic transitions, we infer distances from the current states and next states instead of directly

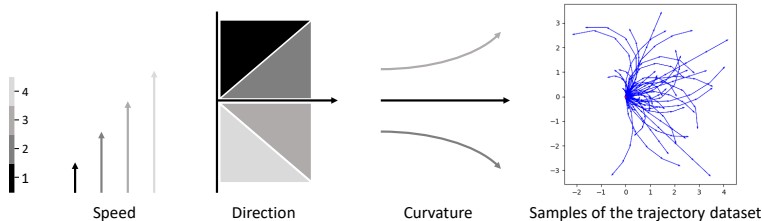

Figure 9: The labeling criteria and the trajectories in the dataset.

predicting next states. Therefore, we feed the distances as state inputs for the encoding LSTM, and the state decoder also outputs mean and variance for the distances. For training losses, we used the negative log-likelihood for the state outputs and the cross-entropy for the label outputs. For activation functions of the output layers, we used linear functions for the mean and log-variance of the state decoder, and softmax functions for the label decoders. In Equation 4, we set the hyperparameters as $\gamma = 10$, $\beta = 9$. We trained the model using Adam optimizer (Kingma & Ba, 2014) with the fixed learning rate of $2e$-4, the batch size of 2048, and 300 epochs.

### A.1.3 POLICY NETWORK

For the policy optimization, we used the SAC algorithm (Haarnoja et al., 2017). To learn the position-agnostic behaviors, we used the state input $s$ to the policy $\pi(a|s, z)$, excluding the positional information of $(x, y)$ coordinates in Ant and $x$ coordinate in Half-Cheetah as in DADS (Sharma et al., 2019). While we trained the WET-VAE with the maximum speed of one, we computed the log-likelihood $\log p_\theta(\tau|z)$ for the reward by reducing the mean by 1/10 and the variance by 1/20 in Equation 6, considering the allowable length of one-step movement of the Ant. For Half-Cheetah, we reduced them by 1/5 and 1/20, respectively. In the SAC, we used the entropy temperature of 0.01. We trained the model using Adam optimizer (Kingma & Ba, 2014) with the fixed learning rate of $3e$-4, the batch size of 256, and 64 iterations per epoch for totally 30000 epochs. For other hyperparameters, we used the discount factor of 0.99, target smoothing coefficient of 0.01, the latent sampling distribution as the unit Gaussian distribution, and two hidden layers of $(512, 512)$ for all of the actor, critic and value function in Ant, and $(256, 256)$ in Half-Cheetah.

### A.1.4 PLANNING

We set primitive horizons $H_z = 20$ for DADS and $H_z = 40$ for WEDIS and WEDIS-S, which are determined by comparing the average speed. For SeCTAR and WEDIS-NTS, we set trajectory lengths as $T = 40$ and $T = 20$, respectively. According to the primitive horizons and trajectory lengths, we set episode horizons as $H_E = 300$ for DADS and $H_E = 600$ for others. For dense reward tasks, we used planning horizons $H_P = 4$ for all methods except for $H_P = 3$ of WEDIS. In contrast, for sparse reward tasks we used the maximum planning horizons $H_E/(H_z \times T)$ for all methods. These planning parameters are summarized in Table 2.

For sampling distribution of the skills, DADS that was trained from the uniform distribution used a uniform distribution from -1 to 1, and others that were trained from the normal Gaussian distribution used a uniform distribution from -1.5 to 1.5 except for the dimension of speed factor of WEDIS and its ablations. For WEDIS and the ablations, we limit the minimum speed as the one sampled from zero since too low speed is useless for the navigation. Note that this strategy is also an advantage of our method thanks to the disentanglement and interpretability of the skills. For the predictions, we used the mean output from the WET-VAE decoder as the trajectories without sampling. The actions of an agent were sampled from the distributions given by the policy.

We used the collision penalty weight $w_p = 100$, the reward weight $w_r = 5$, and the acceptable error $\epsilon = 5$ in the sparse reward cases. In Figure 6, the goals are positioned on the Cartesian coordinate of $(24, 12)$. As the straight paths are blocked on the walls in the mazes, the effective distance to goals is 36.

| Algorithms | $H_E$ | $H_z$ | $T$ | Steps / Skill | $H_P$ (Dense / Sparse) | Sampling Distribution |
|---|---|---|---|---|---|---|
| SeCTAR | 600 | 1 | 40 | 40 | 4 / 15 | $U(-1, 1)$ |
| DADS | 300 | 20 | 1 | 20 | 4 / 15 | $U(-1.5, 1.5)$ |
| WEDIS-S | 600 | 40 | 1 | 40 | 4 / 15 | $U(-1.5, 1.5)$, $U_{speed}(-1.5, 0)$ |
| WEDIS-NTS | 600 | 1 | 20 | 20 | 4 / 30 | $U(-1.5, 1.5)$, $U_{speed}(-1.5, 0)$ |
| WEDIS | 600 | 40 | 5 | 200 | 3 / 3 | $U(-1.5, 1.5)$, $U_{speed}(-1.5, 0)$ |

Table 2: The MPC parameters

## A.2 DERIVATION

### A.2.1 VARIATIONAL LOWER BOUND

$$
\begin{aligned}
\log p_\theta(\tau, y) &= \log \int_z p_\theta(\tau, y, z) dz \\
&= \log \int_z p_\theta(\tau, y|z) p(z) dz \\
&= \log \int_z \frac{p_\theta(\tau, y|z) p(z)}{q_\phi(z|\tau, y)} q_\phi(z|\tau, y) dz \\
&\geq \int_z q_\phi(z|\tau, y) \log \frac{p_\theta(\tau, y|z) p(z)}{q_\phi(z|\tau, y)} dz \\
&= \int_z q_\phi(z|\tau, y) \log p_\theta(\tau, y|z) dz + \int_z q_\phi(z|\tau, y) \log \frac{p(z)}{q_\phi(z|\tau, y)} dz \\
&= \mathbb{E}_{q_\phi(z|\tau, y)}[\log p_\theta(\tau, y|z)] - D_{KL}(q_\phi(z|\tau, y) \| p(z)) = L(\theta, \phi ; \tau, y),
\end{aligned}
\tag{7}
$$

where the inequality is from Jensen's inequality. The log-likelihood $\log p_\theta(\tau, y|z)$ is decomposed as below.

$$
\begin{aligned}
\log p_\theta(\tau, y|z) &= \log \left[ p_\theta(\tau|z) p_\theta(y|z) \right] \\
&= \log \left[ p_\theta(s_1, ..., s_T|z) p_\theta(y^1, ..., y^M|z) \right] \\
&= \log \left[ p_\theta(s_1|z) p_\theta(s_2|s_1, z) \cdots p_\theta(s_T|s_{1:T-1}, z) p_\theta(y^1|z) \cdots p_\theta(y^M|z) \right] \\
&= \log \left[ \prod_{t=1}^{T} p_\theta(s_t|s_{1:t-1}, z) \prod_{m=1}^{M} p_\theta(y^m|z) \right] \\
&= \sum_{t=1}^{T} \log p_\theta(s_t|s_{1:t-1}, z) + \sum_{m=1}^{M} \log p_\theta(y^m|z)
\end{aligned}
\tag{8}
$$

Therefore, the objective function of WET-VAE becomes

$$
\begin{aligned}
L(\theta, \phi ; \tau, y) &= \mathbb{E}_{q_\phi(z|\tau, y)}[\log p_\theta(\tau, y|z)] - D_{KL}(q_\phi(z|\tau, y) \| p(z)) \\
&= \mathbb{E}_{q_\phi(z|\tau, y)}\left[\sum_{t=1}^{T} \log p_\theta(s_t|s_{1:t-1}, z) + \sum_{m=1}^{M} \log p_\theta(y^m|z)\right] - D_{KL}(q_\phi(z|\tau, y) \| p(z))
\end{aligned}
\tag{9}
$$

By adding the additional weights to this equation, we finally obtain Equation 4.

### A.2.2 POLICY REWARD

$$
\begin{aligned}
\mathbb{E}_{p(z)}[D_{KL}(p_\psi(\tau|z)\|p_\theta(\tau|z))] &= \mathbb{E}_{p(z)}[\int_\tau p_\psi(\tau|z)\log\frac{p_\psi(\tau|z)}{p_\theta(\tau|z)}\,d\tau] \\
&= \int_z p(z)\int_\tau p_\psi(\tau|z)\log\frac{p_\psi(\tau|z)}{p_\theta(\tau|z)}\,d\tau\,dz \\
&= \int_z\int_\tau p(z)p_\psi(\tau|z)\log\frac{p_\psi(\tau|z)}{p_\theta(\tau|z)}\,d\tau\,dz \\
&= \int_z\int_\tau p(z)p_\psi(\tau|z)[\log p_\psi(\tau|z) - \log p_\theta(\tau|z)]\,d\tau\,dz \\
&= \int_z\int_\tau p(z)p_\psi(\tau|z)\log p_\psi(\tau|z)\,d\tau\,dz - \int_z\int_\tau p(z)p_\psi(\tau|z)\log p_\theta(\tau|z)\,d\tau\,dz \\
&= \int_z p(z)\int_\tau p_\psi(\tau|z)\log p_\psi(\tau|z)\,d\tau\,dz - \int_z\int_\tau p_\psi(\tau,z)\log p_\theta(\tau|z)\,d\tau\,dz \\
&= -\mathcal{H}(p_\psi(\tau|z)) - \mathbb{E}_{p_\psi(\tau,z)}[\log p_\theta(\tau|z)]
\end{aligned}
\tag{10}
$$

As the above equation is an objective function for a minimization problem, we can obtain Equation 6 by changing the sign.

### A.3 ADDITIONAL EXPERIMENTAL RESULTS

### A.3.1 WET-VAE

In this section, we first check the effects of the two weights, $\beta$ and $\gamma$ in Equation 4. Then, we show the results of the three additional experiments to learn other representations; 1) learning polar coordinate representation and Cartesian coordinate representation using proper weak labels for each representation in 2D. 2) disentanglement of speed and acceleration in 1D. 3) disentanglement of four factors of speed, direction, curvature and acceleration in 2D. For each of the three additional experiments, we generated the trajectory dataset again by following Table 1.

Each column in Figure 10 shows the latent traversals of models with all the same hyper-parameters except for $\beta$ and $\gamma$. All models are trained with same dataset used in Section 6.1.1. For the model in (e), we did not reuse the model in Section 6.1.1, but reproduced it again for this experiment. These results imply that the proper choices of both $\beta$ and $\gamma$ are required for the training of the WET-VAE.

Figure 11 shows the latent traversals of representations of a two-dimensional latent variable $z \in \mathbb{R}^2$ learned with the polar coordinate and Cartesian coordinate. For the polar coordinate representation, we used weak labels of radius divided into four segments with 0.25 interval from zero to one, and angle divided into four segments with an interval of 90 degrees from zero to 360 degrees. For the Cartesian coordinate representation, we used weak labels of $x$ and $y$ coordinates divided into four segments with 0.5 interval from -1 to 1. For comparison, we also provide the results of learning Cartesian coordinate in an unsupervised manner without weak labels. The three models in Figure 11 are trained with the same trajectory dataset generated by the combination of the radius (speed) and angle (direction). It is worthy of noting that the uses of different ($x$ and $y$ coordinates) or no labels can result in different representations from the true generative factors (radius and angle). This implies the importance of the inductive bias to learn the desired factors.

To disentangle speed and acceleration factors in 1D, we set the weak labels divided into four segments from -1 to 1 for speed, and three segments of zero, positive and negative values the acceleration. Figure 12 shows that the WET-VAE learns the disentangled representation of a two-dimensional latent variable $z \in \mathbb{R}^2$ with speed and direction factors in 1D. In this case, the factor contributing to single-step transition $z^{single}$ is the speed factor $z^{speed}$, and the factor contributing over multiple time-steps $z^{multi}$ is the acceleration factor $z^{acc}$. On the other hand, when there is no weak supervision, the model does not learn the meaningful representations.

In Figure 13, it is shown that the WET-VAE can learn the disentangled representations with multiple factors in both $z^{single}$ and $z^{multi}$, where $z^{single} = (z^{speed}, z^{dir})$ and $z^{multi} = (z^{acc}, z^{cur})$. In this

case, the model learns the four-dimensional latent representations $z \in \mathbb{R}^4$ with weak labels for each factor, where we set the full-circle range for the direction.

As shown in Figure 11 and Figure 13, even if the model was trained with dataset of the full-circle angles with the weak supervision, the empty regions appear at some angles. The empty regions were also shown differently at each training even with the same hyperparameters and dataset. In this reason, we trained the model with the semicircle angles in the main experiments to completely cover the only positive directions even though the model trained with full circle angles can cover the larger range of the direction.

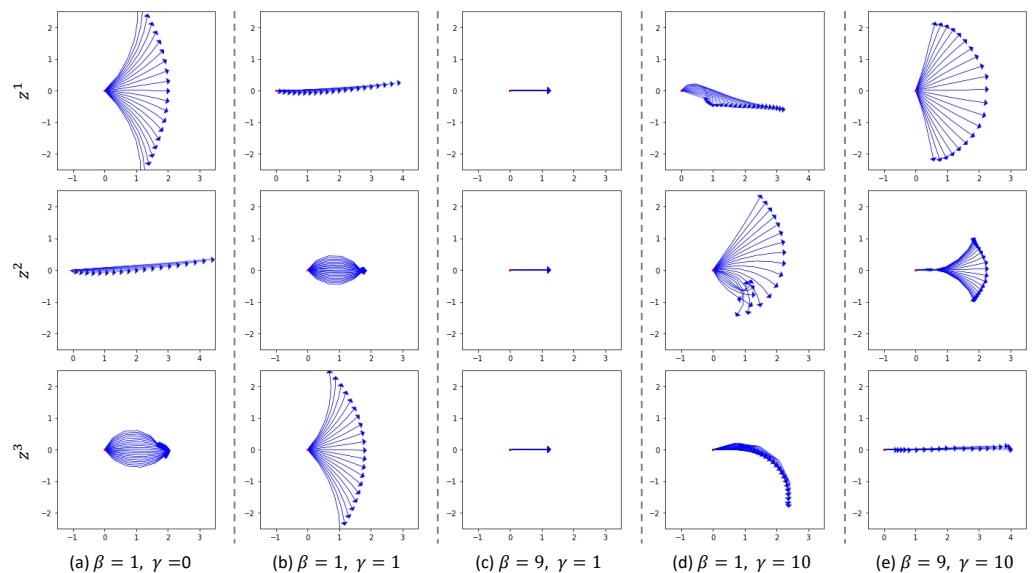

Figure 10: The latent traversals from -1.5 to 1.5. (a) $\sim$ (e) show the results from the models trained with all the same hyper-parameters except for $\beta$ and $\gamma$. The trajectories in all figures are plotted with $T = 5$.

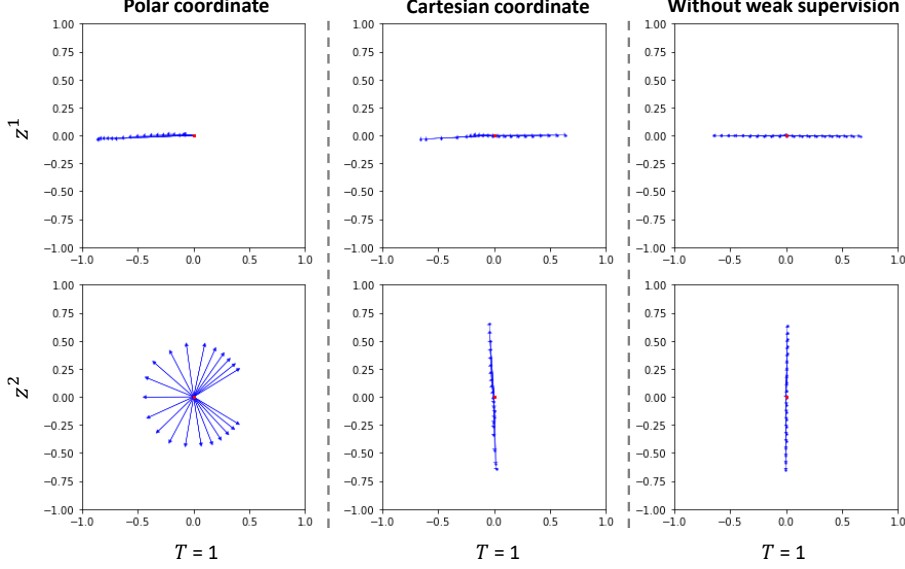

Figure 11: The latent traversals from -1.5 to 1.5 (Left) polar coordinate representation (Middle) Cartesian coordinate representation (Right) representation learned without weak supervision.

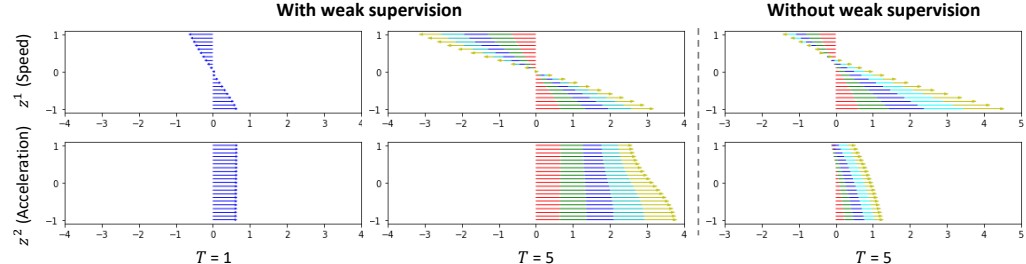

Figure 12: The latent traversals from -1 to 1. WET-VAE can learn the disentangled representations of the acceleration and speed in 1D. For visibility, we represent the values of the latents at the $y$ axis and use different colors at each time-step for $T = 5$. Since the speed is zero for $z^1 = 0$, the first two figures of the second rows are plotted with $z^1 = -1$.

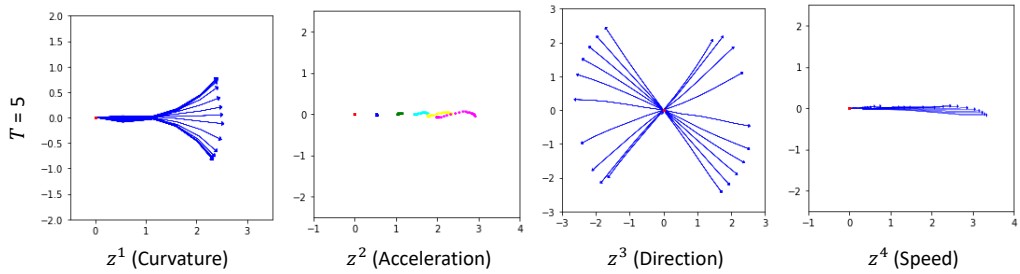

Figure 13: The latent traversals from -1.5 to 1.5. WET-VAE can learn the disentangled representations of the curvature, acceleration, direction and speed. For visibility, we plot the acceleration figure with the points with different colors for each time-step, instead of the arrowed lines.

### A.3.2 POLICY NETWORK ON ANT OF FULL CIRCLE ANGLES AND HALF-CHEETAH

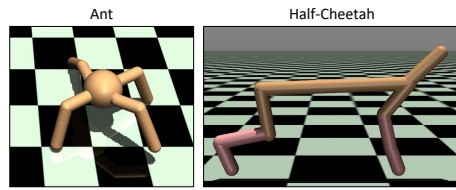

Figure 14: Mujoco environments. (Left) Ant (Right) Half-Cheetah

We present the additional experimental results for the Ant and the Half-Cheetah of Mujoco in Figure 14. Figure 15 shows that the Ant learns the disentangled skills of the speed and direction of the full circle angles in 2D. In Figure 16, it is shown that the Half-Cheetah learns the disentangled speed-acceleration skills in 1D.

To train the policy network for the Half-Cheetah, the 1D speed-acceleration model in Figure 12 is used as the predictive model. After training, the relation function was set as $f^{speed}(z_t^{speed}, z^{acc}) = z_t^{speed} - 0.1 \cdot \frac{z_t^{speed}}{|z_t^{speed}|} \cdot z^{acc}$.

### A.3.3 MORE PLANNING EXAMPLES

We provide the additional MPC results of the main experiment for the Maze1 and Maze 2 in Figure 17. Additionally, we presents more results to show that our method is sample-efficient with the examples in S-shaped maze and obstacles in Figure 18. In each planning procedure, we used a sequence of just two skills for the maze and just a single skill for the obstacle, respectively.

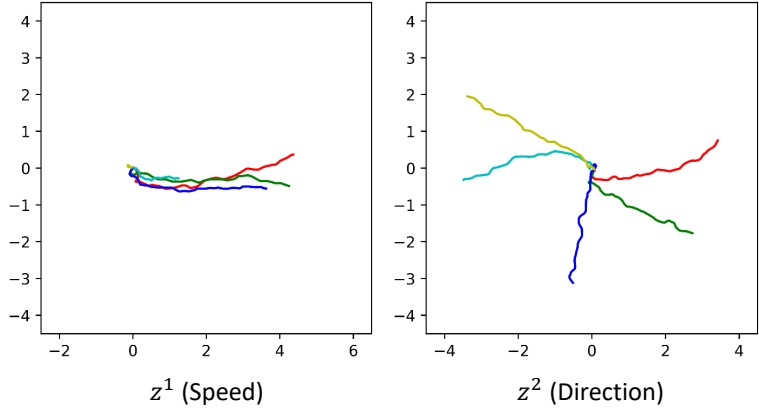

Figure 15: The latent traversals in Ant trained with speed and the direction ranging the full circle angles.

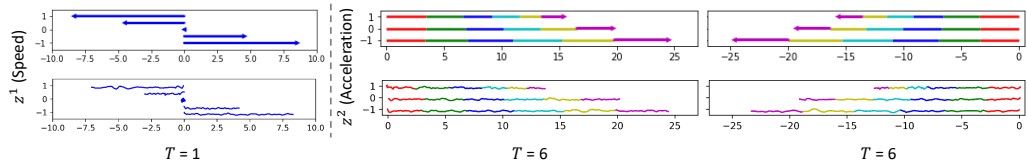

Figure 16: The latent traversals from -1 to 1 in Half-Cheetah. (Top) predicted trajectories by the predictive model. (Bottom) actual trajectories by the Half-Cheetah. For visibility, we represent the values of the latents at the $y$ axis and use different colors at each time-step for $T = 6$.

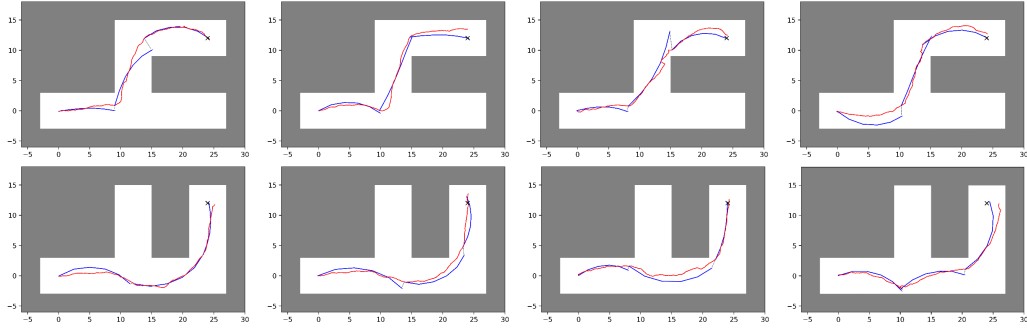

Figure 17: Additional MPC results. (Top) Maze1. (Bottom) Maze2.

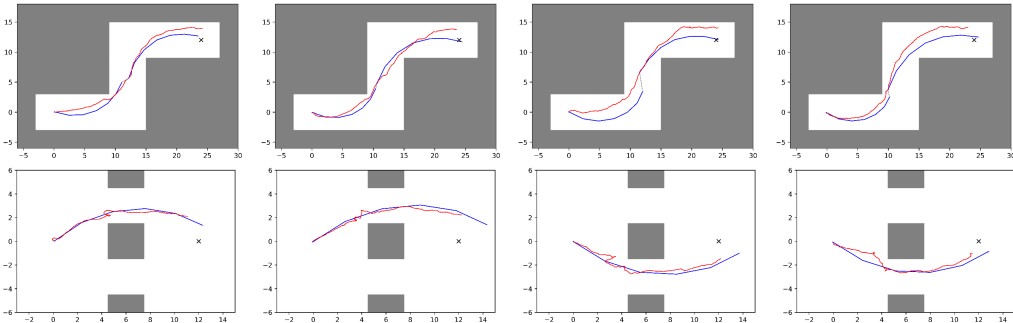

Figure 18: Additional MPC results. (Top) S-shaped maze. We use $T = 6$ and $H_z = 50$ for a sequence of two skills. (Bottom) Obstacles. $T = 5$ and $H_z = 50$ for single skill.

