# OpenReview forum: "Weakly-Supervised Learning of Disentangled and Interpretable Skills for Hierarchical Reinforcement Learning"
_ICLR.cc/2022/Conference — ICLR 2022 Submitted_

### Official Review · Reviewer_CPCV · 2021-11-01

**Correctness:** 3
**Technical Novelty And Significance:** 3
**Empirical Novelty And Significance:** 3
**Recommendation:** 5
**Confidence:** 3

**Main Review:**

My main concerns with the work are its lack of clarity and how many decisions related to the design of the algorithm are not well justified.

- As a non-expert, I found the paper hard to digest. In the process of reviewing this paper, I read some other already published works and found them easier to follow and better motivated. For instance, a clean definition of what is meant by "skill" would have been helpful.
- The weak-labels used to improve disentanglement are computed from the synthetic trajectories. From what I understand, they contain a lot of information about the direction, speed and curvature. If the goal of the WET-VAE is to extract direction, curvature and speed from the trajectories, why not using supervised learning to do that? For instance, one could train a neural network to output direction, speed and curvature using the supervision of the computed labels y. Then, one could use this encoder trained with supervision to train a generative model conditioned on these.
- The advantage of disentanglement is not obvious here (apart from interpretation). It seems to be used only to separate z into z^single and z^multi. But it is not clear if this is useful or why it should be useful. This connects to the following points related to the MPC approach.
- In Algorithm 2 when executing the policy, I did not understand why it was important to exclude z^{multi} from the latent representation fed to the policy. An ablation study would be convincing.
- In Algorithm 2, the role of the hand-crafted functions f is unclear to me. My understanding is the following: when executing a rescaled plan, it is used to update skill z from one rescale step to another. What is the advantage of doing so instead of just using z*_1 across the subtrajectory?
- In Algorithm 2, at each step, planning in the latent space is performed to pick the best action. What is the distribution from which the potential trajectories z_1, …, z_{H_p} are sampled? Is it uniform?
- In some sense, both the artificially generated sequences and the scaling factor H_z must be chosen by the user. It seems H_z is thus redundant since the appropriate sequence length could have been chosen  prior to the sequence generation, no? This question also makes me doubt the usefulness of the hand-crafted functions f, which I suspect are useful only because of the rescaling.



**Summary Of The Paper:**

This work proposes to learn representations of artificially generated trajectories using a VAE-based approach that relies on weak labels to improve disentanglement. Then, a skill-based policy (i.e. a policy conditioned on the learned sequence representation) is trained to imitate the learned model via soft actor-critic. Moreover, the paper introduces a hierarchical planning strategy that explores at the “aggregated sequence level”, i.e. the representation space, and allows for “trajectory rescaling”. The authors show empirically both the disentanglement of WET-VAE and the advantage of their hierarchical approach.

**Summary Of The Review:**

Given the lack of clarity and the multiple algorithm design decisions that are not sufficiently justified, I cannot recommend acceptation. I cannot comment on the novelty of the contribution as I am not well versed into hierarchical reinforcement learning.

---

> ### Author Response · Authors · 2021-11-22
> **Response**
>
> We appreciate the reviewer for the constructive feedback on our paper. We will answer your concerns below.
>
> $\textbf{Q1.}$ We sincerely apologize for making it difficult to understand due to the lack of clarity. In the context of RL, policies performing lower-level tasks or behaviors are referred to as skills or options. Typically, conditioned as an additional input on a variable $z$ that represents each task, those policies perform the tasks. In our work, the skill means the policy that behaves to generate a specific trajectory based on a latent variable $z$. Thank you once again for pointing out the factors that hinder the clarity of the paper.
>
> $\textbf{Q2.}$ The purpose of using VAE architecture is to obtain a probabilistic generative model for trajectories based on a combination of the factors. In the probabilistic generative model, the latent variable z that is considered as a skill is sampled from a prior distribution ${p(z)}$ such as ${z{\sim}p(z)}$. Then, based on the skill, the corresponding trajectory is generated from the learned generative model such that ${\tau{\sim}p(\tau|z)}$. In the VAE architecture, the decoder can be used as the generative model.
>
> $\textbf{Q3.}$ We sincerely apologize for making it difficult to understand due to the lack of clarity. The disentanglement allows for an elaborate control since each factor of a continuous latent variable (skill) can be handled separately. To demonstrate a possible benefit of the disentangled skills with interpretable factors, we provided a planning strategy with the trajectory scaling that makes the planning efficient due to the reduced effective planning horizon. If a skill is entangled, for example, with two factors of speed and direction, this planning strategy is unavailable since we cannot handle those factors separately. In this case, even if we change a value of one dimension of a skill (that is a continuous latent variable), both factors are affected, making it difficult to elaborately control the agent. Also, other than the separation into ${z^{single}}$ and ${z^{multi}}$, all factors included in ${z^{single}}$ and ${z^{multi}}$ are disentangled in one latent variable ${z}$. For example, in the experiment in Section 6.1.1, all three factors (speed, direction, curvature) are disentangled in a three-dimensional latent variable ${z}$. The separation of ${z}$ into ${z^{single}}$  and ${z^{multi}}$  is merely for the policy to be trained more simply and be used more practically with the proposed trajectory scaling.
>
> $\textbf{Q4.}$ The reason to exclude ${z^{multi}}$  from the latent representation fed to the policy is to train the policy on factors for single-step transition. The pre-trained predictive model is reused for reward computation, but for single-step transitions rather than trajectories over multiple steps. That is to say, given a skill, the length of the rollout of the prediction by the predictive model is one. If the rollout is generated for a length greater than one, it becomes a trajectory. While we might train the policy over trajectory-level actions using the RNN architecture, we instead trained over the single-step transition level for simplification of training and practical application such as the proposed trajectory scaling.
>
> $\textbf{Q5.}$ As you understood, when executing a rescaled plan, the hand-crafted function $f$ is used to update skill z from one rescaled step to another. If we do not update the skill for each time-step but hold on $z^*_1$ for all time-steps, the policy will generate a straight line by the size extended by a factor of the number of total time-steps along the first step’s transition,  rather than a corresponding trajectory, e.g., which might be a curve.
>
> $\textbf{Q6.}$ We used a uniform distribution and the details are in Appendix A.1.4.
>
> $\textbf{Q7.}$ As you understood, both the sequence length and scaling factor should be properly chosen for planning and the function f is for the rescaling of the trajectory. The difference between the sequence length and scaling factor in their usages is that the scaling factor is for rescaling the length of transition distances for each time step, whereas the sequence length is the number for which the transitions are performed. For example, given a skill that can generate a semicircle-shaped trajectory for sequence length $T$, we can generate the same shaped (semicircle) trajectory with the size extended by a factor of 2 if we choose $H_z$=2. On the other hand, if we apply the same skill for $2T$ that is twice of the original sequence length, the shape of the generated trajectory will become a full circle.

---

> > ### Comment · Reviewer_CPCV · 2021-11-29
> > **Response**
> >
> > I thank the authors for carefully addressing my concerns and misunderstandings. The explanations provided are reasonable, but I believe some of them could be made more convincing with simple minimalist experiments. For instance, what happens when feeding both z^single and z^multi in the policy? What happens if we keep z fixed without updating it? Unfortunately, I still believe the authors haven't made a convincing case for their approach since many algorithm design decisions are left unjustified. Also, other reviewers have pointed out the fact that the method should be compared to methods using similar amount of supervision. For the above reasons, I will keep my score unchanged.

---

### Official Review · Reviewer_FYox · 2021-11-02

**Correctness:** 3
**Technical Novelty And Significance:** 2
**Empirical Novelty And Significance:** 2
**Recommendation:** 3
**Confidence:** 3

**Main Review:**

Overall, I found the method introduced in the paper to be somewhat complex, and the experimental section to not justifying this complexity. The overall ingredients (VAE on trajectories, train skill policies on a resulting reward signal) are among the current techniques used in skill discovery (e.g., OPAL (https://arxiv.org/abs/2010.13611), EDL (http://arxiv.org/abs/2002.03647)). While the authors show gains over purely unsupervised skill learning methods (DADS, SeCTAR) on the downstream navigation tasks, my main concern is the way that supervision is introduced in WEDIS. I have trouble understanding the motivation for an auto-encoder setup when reference trajectories can be generated beforehand. Why not simply pre-train a policy with a dense reward to explicitly follow these trajectories, conditioned explicitly on the salient factors (direction, speed, curvature)? This should at least be a baseline in the paper. I can see how the training setup in WEDIS could, in principle, allow for more generality, but then again it requires the ability to generate reference trajectories (which in turn would make it possible to train policies to follow them directly). The experiments also lack a comparison to plain Soft Actor-Critic.

Finally, I don't really agree with the premise that current skill learning methods are fundamentally hindered by a lack interpretability. While it's true that the availability of a model enables the usage of standard planning methods like MPC, several works also showed effective hierarchical control with RL, where explicit semantics are not strictly required (e.g., DIAYN, OPAL (https://arxiv.org/abs/2010.13611) or NPMP (https://arxiv.org/abs/1811.11711)).

The writing would benefit from an overhaul wrt grammar.

**Summary Of The Paper:**

The paper at hand proposes a new framework for pre-training skill policies (WEDIS), and use them for control in a hierarchical setup with MPC. The main idea is that skill policies should follow a set of generated trajectories based on some salient factors. For this, the authors train a VAE that will then provide both the control variables (latent encoding) and a predictive model of the trajectory (decoder). The skill policy is trained to match the trajectories of the predictive model, and the predictive model is then used for MPC.

**Summary Of The Review:**

In my view, the experiments in the paper fail to justify the complexity of the method that was introduced. With the amount of supervision required, simpler and potentially more effective approaches come to mind but are missing in the comparison.

---

> ### Author Response · Authors · 2021-11-22
> **Response**
>
> We appreciate the reviewer for the constructive feedback on our paper. We will answer your concerns below.
>
> $\textbf{Q1. I have trouble understanding the motivation for an auto-encoder setup when reference trajectories can be generated beforehand.}$
> $\textbf{Why not simply pre-train a policy with a dense reward to explicitly follow these trajectories, conditioned explicitly on the salient factors}$
> $\textbf{(direction, speed, curvature)?}$
>
> : We thank the reviewer for the recommendation of the relevant papers and constructive feedback. Our method basically is motivated by the SeCTAR framework. In this framework using the variation auto-encoder (VAE) setup, collected trajectories are embedded into latent space by the encoder of VAE. Then, the decoder reconstructs the trajectories based on latent variables, where the decoder is used for the skill-based predictive model for the policy. Our main goal is to make the policy follow the predictive model based on the same skills that consist of disentangled factors so that the skills can be predictable on trajectory-level for effective planning. To this end, we also adopted the VAE, where the decoder can be used as the predictive model as well as reward computer to train the policy. Similar to the existing methods such as SeCTAR and DADS, as our predictive model enables the policy to solve downstream tasks with zero-shot via model-based planning, there is no need to additionally train higher-level policies for each downstream task. Therefore, our method differs from pre-training the policy by dense rewards without the predictive model, since this requires training the higher-level policy for each downstream task.
>
> $\textbf{Q2. I don't really agree with the premise that current skill learning methods are fundamentally hindered by a lack interpretability}$
>
> : We sincerely apologize for making it difficult to understand due to the lack of clarity, and thank for the recommendation of relevant papers. We agree that the claim is too strong, and thus re-express as “the disentanglement can enhance the applicability of skills” instead of “a lack of the disentanglement limits their applicability in the previous methods”.
>
> $\textbf{Q3. The writing would benefit from an overhaul wrt grammar.}$
>
> : We thank the reviewer for the comments.

---

### Official Review · Reviewer_BoDo · 2021-11-03

**Correctness:** 2
**Technical Novelty And Significance:** 2
**Empirical Novelty And Significance:** 2
**Recommendation:** 3
**Confidence:** 4

**Main Review:**

## Strengths
- The related work section provides a comprehensive overview of relevant prior works.

- The provided planning visualizations clearly show the differences between the learned skill spaces and how they influence downstream learning.

- The appendix provides detailed instructions for generating the used dataset and reproducing the policy training.


## Weaknesses
I have a number of major concerns with respect to the proposed problem formulation and approach:

1. The proposed approach is compared to unsupervised skill learning approaches (like DIAYN, SeCTAR, DADS), but it assumes access to an automatically pre-generated dataset of training trajectories. This directly violates the premise of unsupervised skill-*discovery* approaches: if we were able to generate meaningful trajectories in the first place, we would not need to train an agent to discover them. This becomes more apparent if we move from simple 2D ant navigation problems as used in the paper's evaluation to more complex problems like kitchen manipulation with a robot arm: there it would be very challenging to pre-define object interaction primitives etc, which is why unsupervised skill discovery approaches were proposed in the first place. Thus, I believe the problem formulation of the paper is fundamentally flawed.

2. The proposed approach does not learn disentangled skills with "weak supervision" but assumes fully annotated training trajectories with all relevant factors of variation given for every time step. Thus I see the proposed method as a fully supervised approach to learning a disentangled latent space instead with strong assumptions on the available supervision. Defining this set of "interesting factors of variation" in the first place requires considerable human effort and can be non-trivial in harder problems.

3. The paper does not motivate clearly *why* it would be beneficial to train a latent space with disentanglement of the given factors of variation. This should be much clearer motivated in the introduction. Currently it only vaguely mentions that a lack of disentanglement and interpretability of skills "limits their applicability".

4. The explanation of the model is in part unclear. I am specifically confused about section 4.2.2 (see questions below). This also makes understanding the downstream usage of the model challenging.

5. Section 5 introduces an approach which the paper calls "trajectory scaling". From my understanding it simply describes the common RL practice of "action repeat" which is eg used by default in many Atari games. Although it is less common in MPC-based planning approaches this connection should be made in the paper.

6. The experimental comparisons to *truly* unsupervised approaches like SeCTAR and DADS are not fair in my understanding: these approaches do not have access to a pre-collected dataset of demonstrations and thus need to learn to explore the environment while learning the skill policy. This is a much harder task, but this distinction is never acknowledged in the experimental section.

7. The writing of the paper in part impairs its clarity. Many formulations stay vague or are not put in appropriate context, eg from the abstract without any further explanation of "single-step transitions" or "trajectory-level behaviors": "Additionally, we propose to train a policy network with single- step transitions and perform the trajectory-level behaviors at test time with the knowledge on the skills, which simplifies the exploration problem in the training." In other instances the writing does not seem to explain all the details necessary to understand the introduced concepts (see point (4) above).


## Questions
- I am confused why we need to train single-step transition models for reward computation instead of training with the pre-trained trajectory-level models as reward-computers. Is this to get a denser reward function? How does the model in Figure 1 change when we split the latent variable into z_single and z_multi? How does "heuristically" finding the relation functions between z_single and z_multi work exactly?


**Summary Of The Paper:**

The paper proposes an approach for learning interpretable skill embeddings by leveraging an automatically generated dataset of trajectories annotated with factors of variation like direction, speed and curvature which should be used to structure the learned skill space. It shows that these factors can indeed be used to learn a latent space with disentangled factors of variation which can be used via an MPC-based planning method for solving downstream tasks in a OpenAI Gym ant environment.

**Summary Of The Review:**

In my opinion the assumptions made in the paper contradict the premise of the unsupervised skill discovery problem. Further, the approach is not clearly described, the experimental evaluation is not fair due to differing assumptions and the overall writing of the paper needs improvement. Thus, I do not recommend acceptance of the submission.

---

> ### Author Response · Authors · 2021-11-22
> **Response related to Questions**
>
> $\textbf{Q1. I am confused why we need to train single-step transition models for reward computation instead of training with the pre-trained}$
> $\textbf{trajectory-level models as reward-computers. Is this to get a denser reward function? How does the model in Figure 1 change when}$
> $\textbf{we split the latent variable into ${z^{single}}$ and ${ z^{multi}}$?}$
>
> :  The pre-trained trajectory level model is the same as the model for reward computation. The pre-trained model is reused for reward computation, but for single-step transitions rather than trajectories over multiple steps. That is to say, given a skill, the length of the rollout of the prediction by the predictive model is one. If the rollout is generated for a length greater than one, it becomes a trajectory that is a sequence of transitions. While we might train the policy for trajectory-level actions using the RNN architecture, we instead trained the feedforward policy over the single-step transition level for simplification of training and practical application such as the proposed trajectory scaling.
>
> $\textbf{Q2. How does "heuristically" finding the relation functions between ${z^{single}}$ and ${z^{multi}}$ work exactly?}$
>
> :  The reason to find the relation function heuristically is that the factors are embedded in latent space. Consider two factors of direction and curvature (difference between angles of two time-steps) for example as in the experiment, where the direction is for single-step transition and the curvature is for multiple-steps. Conditioned on a latent variable sampled from prior distribution ${z=[z_1,z_2 ]= [z^{dir},z^{cur}] {\sim} p(z)}$ (the permutation can be changed), the decoder of WET-VAE that is used as a predictive model generates the corresponding trajectory, which can be denoted as ${\tau=[s_1,s_2,…s_T ]=[D_1(z),D_2(z),…,D_T(z)]}$, where ${D_t(z)}$ denotes the mean output of decoder for ${t^{th}}$ rollout with respect to z. In contrast, we trained the policy to generate the same trajectories for single-steps with the predictive model, where the policy takes an action by conditioned on z^dir and generates a trajectory that is the same as ${s_1= D_1 ([z^{dir},0])}$ for the first-step transition (as ${z^{cur}}$ has no effect on the first step, it does not matter what value is used for ${z^{cur}}$ but we used zero for convenience without loss of generality). When generating a trajectory for multiple time-steps, we can achieve it by changing the direction factor at each time step such that ${\tau'}$ =${[s'_1,s'_2,…s'_T]}$=${[D_1([z^{dir},0]), D_1([z^{dir}+weight {\times} z^{cur},0]), …, D_1([z^{dir}+(T-1) {\times} weight{\times}z^{cur},0])]}$. Then, we set a relation function for direction and curvature as ${f(z^{dir}, z^{cur}) = z^{dir}+weight{\times}z^{cur}}$ and find the weight value so that $\{\tau = \tau’}$. For simplicity, we did not use any optimization technique, but heuristically found an appropriate weight that produces the most similar tau’ when inserting several values into the weight.

---

> > ### Comment · Reviewer_BoDo · 2021-11-27
> > **Rebuttal Reply**
> >
> > Thanks for your reply! I agree that skill discovery approaches like SeCTAR and DADS are not natural baselines for the proposed approach. I think when I read the paper I was confused by the fact that the main experimental comparison was done with respect to these approaches -- I agree with the authors that the writing should be adjusted to address this point.
> >
> > However, the main issue of the submission remains: while a lot of supervision is provided to the proposed approach in terms of the pre-generated, labeled dataset of expert trajectories, there is no experimental comparison to approaches that are given the same amount of supervision. This was also noted by other reviewers as a major weakness of the submission and the rebuttal did not address this. Thus, I stand by my original evaluation and cannot recommend acceptance for the submission in its current form.

---

> ### Author Response · Authors · 2021-11-22
> **Response related to Concerns (Part 2)**
>
> $\textbf{W2}$. We used the term “weakly-supervised” learning in that the labels are weakly assigned. As you pointed out, it might not be a straightforward work to pre-define the interesting factors of variations, which can be seen as a weakness of our work. Nevertheless, once the factors are defined, the labels can be assigned automatically without manual labeling since they are weak labels.
>
> $\textbf{W3}$.  We sincerely apologize for making it difficult to understand due to the lack of clarity and thank for the comments. We agree that the claim is too strong, thus re-express as “the disentanglement can enhance the applicability of skills” instead of “a lack of the disentanglement limits their applicability in the previous methods”.
>
> $\textbf{W4}$.We sincerely apologize for making it difficult to understand due to the lack of clarity.
>
> $\textbf{W5}$.We thank the reviewer for the suggestion. The proposed trajectory scaling is similar to action repeat in Atari in that the same action is repeated for several frames (time-steps). We used the term “trajectory scaling” in order to indicate that the whole trajectory is extended rather than a transition of a single time-step.
>
> $\textbf{W6}$. As answered in Weakness 1, we proposed to use the synthetic trajectory dataset so that we overcome the difficult exploration task. As you suggested, we will clearly explain the problem statement and motivation in both method and experiment sections.
>
> $\textbf{W7}$. We sincerely apologize for making it difficult to understand due to the lack of clarity. Thank you once again for pointing out the factors that hinder the clarity of the paper.

---

> ### Author Response · Authors · 2021-11-22
> **Response related to Concerns (Part 1)**
>
> We appreciate the reviewer for the detailed and constructive feedback on our paper. We fully acknowledge your comments but will answer your concerns below.
>
> $\textbf{W1}.$
> We highly appreciate the reviewer for the critical and valuable comments about the motivation of this paper. We would like to point out that our contribution lies on learning predictable skills for effective trajectory-level planning rather than discovering a variety of skills in the unsupervised skill learning literature. As you pointed out, the unsupervised skill learning methods are for discovering a variety of skills in an unsupervised manner. Those methods are typically based on information-theoretic rewards to discover and learn the skills. To discover various skills, an agent needs to visit various states or trajectories because the skills can be identified as the distributions for the visited states. This thus essentially requires an effective exploration strategy in the environment, which is an important but difficult issue that must be handled for the unsupervised skill learning approaches.
>
> However, instead of the aspect of discovering of the skills, we focused on the aspect to combine the model-free and model-based methods in which both the predictive model and policy are governed by the same skills. While the existing methods, SeCTAR and DADS, attempt to simultaneously discover and learn skills for both the predictive model and policy in an unsupervised manner, the discovered skills are entangled as shown in the experiments in Section 6.1.2 of this paper when they are continuous latent variables. At the expense of discovering, we proposed to learn disentangled skills consisting of interpretable factors. As a possible benefit of the interpretability, we provided the trajectory scaling method when performing planning. To learn disentangled representations from dataset that is a set of trajectories in our case, the dataset should have learnable factors of variation being capable of generating the data. However, when obtained by an exploration to visit just various states, the trajectory that is a sequence of state’s transitions might consist of irregular transitions without learnable factors. Moreover, the exploration is a difficult problem itself as mentioned before.
>
> In order to alleviate those difficulties, we proposed our framework. Instead of exploring the environment to discover skills, we pre-generate the synthetic trajectories with learnable factors. As the trajectories are generated at a semantic level, such as transitions of positions rather than the full states in the environment, it does not require the knowledge on the environment or training of expert policies as in imitation learning. Although this method might miss the chance to discover novel skills via exploration, it can bypass the difficulty of the exploration problem and still learn useful skills. As you pointed out, those skills can be somewhat limited to pre-defined factors, and pre-defining those factors might not be straightforward. Nevertheless, as our focus is to learn the disentangled skills that can be predictable on trajectory-level for effective planning over the latent space rather than to discover novel skills, we compared with the existing methods such as SeCTAR and DADS because they contribute to controlling the policy over the latent space by the skill-based predictive model in addition to the unsupervised skill discovering.
>
> Unfortunately, in this paper, we agree that this claim is not highlighted throughout the paper and we will fully revise the manuscript to better explain our problem statement and motivation.

---

### Official Review · Reviewer_onQA · 2021-11-03

**Correctness:** 3
**Technical Novelty And Significance:** 2
**Empirical Novelty And Significance:** 3
**Recommendation:** 5
**Confidence:** 4

**Main Review:**

Strengths:

* Interesting visualizations of learned interpretable skills on the Ant navigation task. Additionally, the paper showed that combining interpretable skills with model-based planning (MPC) for long-horizon tasks can result in better performance on maze navigation.

Weaknesses:

(1) The data requirements for the proposed method seem impractical for most real applications. The method requires a synthetically generated dataset (in their experiment, they used 100K trajectories), which are generated by varying the generative factors. If I understood the dataset generation procedure correctly, generating this dataset (of trajectories with varying speeds/directions/acceleration) requires that you already have expert policies that can run at various speeds/directions/accelerations.

1a) How does the dataset size & quality affect the quality of the learned disentanglement, and performance of the learned policy?

1b) (Clarification) Does the 100K trajectory dataset consist of the full observations from Ant/Half-Cheetah (e.g., including joint velocities)? If so, were expert policies and a simulation environment needed to synthetically generate this dataset?

1c) (Clarification) Did the other methods (SeCTAR, DADS) make use of the same 100K trajectory dataset (for fairer comparison), and if so, how?

(2) The idea of using weakly-supervised disentangled representations in order to learn an “interpretable” policy behavior, and comparing against unsupervised representation learning methods such as VAE, has also already been studied in a prior work [1], which was not discussed/cited in the paper. The method in [1] also learns disentangled latent encodings Z, which is used to condition the policy, leading to “interpretable” skills.

[1] “Weakly-Supervised RL for Controllable Behavior”, NeurIPS 2020 https://arxiv.org/abs/2004.02860

(3) In general, I felt that the writing & presentation could be improved.

3a) I had trouble understanding what “latent traversals of trajectories of the decoder” mean (in Figures 4, 5, 10, 11, 13, 15, 16). For example, in Figure 4, could you clarify what the X- and Y-axis labels are, and what each line represents? I am guessing that each line corresponds to a different latent variable (by varying one dimension, while keeping the rest fixed), and the X/Y axes are positions of the ant?

3b) Could you elaborate on *why* the disentanglement helps with MPC planning & result in higher performance (Fig 7)? Could it possibly be because the learned skills are more distinguishable from one another? Can this be verified empirically somehow?

Minor typos:
Page 3: “decesion" -> “decision”
Page 3: “We will focus on the aspect of the generative model of the decoder”: I wasn’t sure what this sentence means. Did you perhaps mean: “we will focus on the generative modelling aspect of the decoder”?
Page 3: “which learns the representations in the unsupervised manner” ->  “which learns the representations in an unsupervised manner”
Page 5: “in the Equation 6” -> “in Equation 6”
Page 7: “based on the each factor” -> “based on each factor”
Figure 3: “solving the long-horizon problems” -> “solving long-horizon problems”
Figure 15: “the direction ranging the full circle angles” could be better worded.
Figure 16: “half-cheetah” -> “Half-Cheetah”


**Summary Of The Paper:**

The paper proposes a method that learns disentangled skill representations, and shows qualitative & quantitative results on Mujoco Ant navigation. (1) They first synthetically generate a trajectory dataset by the combination of different factors. (2) They train a trajectory VAE (Co-Reyes ‘18) that enforces the learned trajectory representations to be disentangled using weak labels (Margonis ‘20). (3) Then, keeping the pre-trained trajectory VAE fixed, they learn a skill-based policy to generate similar trajectories to the learned decoder of the trajectory VAE by minimizing the KL divergence between the trajectory distributions.

**Summary Of The Review:**

My main concern is regarding the practicality of the proposed method (see Weakness #1 above). I also feel that discussion of prior work and overall paper presentation could be improved.

---

> ### Author Response · Authors · 2021-11-22
> **Response**
>
> We appreciate the reviewer for the constructive feedback on our paper. We will answer your concerns below.
>
> $\mathbf{Q1a}$.
> We thank the reviewer for the constructive suggestion. We additionally tested experiments for fewer numbers of trajectories and noisy trajectories, and then checked the latent traversals of the decoder.
> First, we trained the WET-VAE network by reducing the number of datasets from 10k to 1k. When using fewer trajectory data, more training epochs were required and the quality of the learned decoder became slightly poor.
> Second, for the noisy condition, we added Gaussian noises into each transition, where the noise is with zero mean and different variances of 0.1 and 0.01. For the relatively small noise with 0.01 variance, the quality of the learned decoder was almost the same. For larger noises with 0.1 variances, although the quality became poor, it was found that the learning of disentanglement is still valid.
>
> $\mathbf{Q1b}$.
> As described in Section 6.1.2 and Appendix A.1.1, the synthetic dataset consists of just x-y coordinates and does not include the full observations of the environments such as joint’s positions or speeds. Therefore, to generate the dataset synthetically, there is no need for an expert policy or simulation environment. As you pointed out, it is very hard to obtain the offline training dataset for the full observation via generation from hard-coding in a simulation environment or demonstration from an expert policy. Our work also used the hard-coded method for generating the synthetic trajectories, but the trajectories are generated on the semantic level, for example just a sequence of the positions instated of full states. Because this does not require specific knowledge on the environments, our data generation method is much simpler than the generation of the full states or training of the expert policy.
>
> $\mathbf{Q1c}$.
> The other methods used for comparison did not use the synthetic dataset but collected the samples of trajectories through exploration in the environments. The processes of generating and using the training dataset to learn representations of trajectory, instead of collecting it by exploration, are also the contribution of our framework, so we do not think that it is unfair. To be more specific, the existing methods (SeCTAR, DADS) attempt to explore online for collecting the samples of trajectories, often generating meaningless samples (e.g. random walks). To overcome this issue, our method pre-generates the trajectory samples with x-y coordinates which are easy to obtain and then extracts the skills from them by using the proposed trajectory autoencoder (WET-VAE).
>
> $\mathbf{Q2}$.
> We thank the reviewer for the recommendation of the relevant paper. This paper used a weakly-supervised learning method to learn the disentangled representation of single observations such as images. Ours differs from this work in that our method learns the disentangled representation of trajectories that are sequences of states instead of single observations. We will cite this paper with detailed explanations.
>
> $\mathbf{Q3a}$.
> We sincerely apologize for making it difficult to understand due to the lack of clarity. In figure4, the X and Y axes mean x and y coordinates, and each line means the generated trajectory for each latent variable.
>
> $\mathbf{Q3b}$.
> As a possible benefit of the disentangled skills with interpretable factors, we provided a planning strategy with the trajectory scaling that makes planning efficient due to the reduced effective planning horizon. If a skill is entangled, for example, with two factors of speed and direction, this planning strategy is unavailable since we cannot handle those factors separately. In this case, even when we change a value of one dimension of a skill (that is a continuous latent variable), both factors are affected, making it difficult to elaborately control the agent.
>
> $\mathbf{Minor{\ }typos}$. We sincerely apologize for making it difficult to understand due to the lack of clarity. The sentence “We will focus on the aspect of the generative model of the decoder” means that we will focus on a function of the decoder as a generative model. Of the two key functions of VAE: 1. Encoder to embed data into a latent space, ${z{\sim}q(z|x)}$ 2. Decoder to generate data from a latent variable, ${x{\sim}p(x|z)}$ where  ${z{\sim}p(z)}$. After training WET-VAE, we used only the decoder part as a predictive model for a policy. Thank you once again for pointing out this ambiguous sentence.

---

### Decision · Program_Chairs · 2022-01-20

**Decision:**

Reject

**Comment:**

The authors present a method for learning disentangled skill representation that uses weak supervision. The reviewers mentioned that the paper tackles an important problem, delivers interesting and novel visualizations of the learned skills, and positions the paper well in the context of related work. The reviewers also point out several points of criticism: the complexity of the method, lack of convincing comparisons to baselines that utilize the same amount of data and the quality of writing, among others. I encourage the authors to address these points in the future version of the paper.